# PARP1 and PARG Are the Draft Horses for Polycomb-Trithorax Chromatin Regulator Machinery

**DOI:** 10.3390/biom15091314

**Published:** 2025-09-12

**Authors:** Guillaume Bordet, Alexei V. Tulin

**Affiliations:** Department of Biomedical Sciences, School of Medicine and Health Sciences, University of North Dakota, Grand Forks, ND 58202, USA; guillaume.bordet@und.edu

**Keywords:** PARP-1, genetic landscape, chromatin loosening, Polycomb complexes, Trithorax complexes

## Abstract

During tissue differentiation, gene expression patterns are committed to the epigenetic cellular memory machinery, including Polycomb and Trithorax groups (PcG and TrxG), which label chromatin with repressive or active histone marks. Histone marks recruit effector proteins that then execute local chromatin repression or activation. The effectors of TrxG have remained largely unknown. Here we report that the Poly (ADP-ribose) Polymerase 1 (PARP1) and Poly (ADP-ribose) Glycohydrolase (PARG) function as critical effectors of TrxG and PcG, respectively. We found that PARP1 binds TrxG-generated histone marks with high affinity in vitro, completely colocalizing with them genome-wide, and controls the expression of loci modified by TrxG. Conversely, PARG preferentially associates with PcG-occupied loci. We propose a model in which TrxG complexes prime chromatin for PARP1 recruitment, leading to poly (ADP-ribose) generation to maintain an open chromatin state essential for transcription.

## 1. Introduction

Precise activation and silencing of specific genes is the keystone of developmental homeostasis [1]. Throughout this process, chromatin factors mark regions for activation or repression, yet the underlying mechanisms remain unclear. We do know that Polycomb group (PcG) and Trithorax group (TrxG) protein complexes are key players in this regulatory system in that they are traditionally associated with the repression and activation of developmental genes, respectively [2,3] (Figure 1A). Recent research has revealed a more nuanced understanding of these complexes, particularly PcG, which is now recognized for its roles in both gene repression and activation [2,4]. PcG and TrxG complexes function as cellular memory machinery, maintaining chromatin states and orchestrating the placement of transcription factors across cell divisions throughout development [3,5,6,7]. These complexes do not directly modify chromatin structure; instead, they deposit specific histone marks. The mechanism by which TrxG-deposited marks trigger and maintain chromatin opening, while PcG-deposited marks induce and sustain chromatin condensation, has long been a subject of investigation. To explain this phenomenon, the histone modification code hypothesis holds that histone codes serve as patterns of chemical modifications recognized by specialized effector machinery responsible for chromatin opening or condensation [8]. Recent studies have identified several PcG effector proteins, providing insights into Polycomb-mediated repression mechanisms [9,10]. However, the primary effector proteins for TrxG-mediated activation and maintenance of the active state remain elusive. While SWI/SNF (SWItch/Sucrose Non-Fermentable) complexes can facilitate local chromatin remodeling by sliding nucleosomes along DNA, they cannot account for the extensive chromatin opening observed during developmental transitions [11].

As a key regulator of chromatin state involved in both activation and maintenance of the active state, Poly (ADP-ribose) Polymerase 1 (PARP1) is a strong candidate for the role of TrxG effector [12,13,14] (Figure 1B). Upon activation, PARP1 synthesizes poly (ADP-ribose) chains, leading to chromatin opening [14,15]. In *Drosophila*, PARP1 is essential for chromatin loosening and transcriptional activation on steroid-, NFkB-dependent, and stress-activated loci during all developmental stages [14,16,17]. PARP1 also undergoes rapid automodification upon activation and spreads along the gene body, ensuring chromatin loosening in these regions [17,18]. While the precise mechanism that controls PARP1 spreading and binding along an activated locus is unknown, its affinity for histone [19,20] suggests that specific histone marks might facilitate this process.

Accordingly, among all possible histone modifications, we herein demonstrate that PARP1 binds with the highest affinity, both in vitro and in vivo, to histone marks associated with TrxG and, as such, controls the activity of developmental loci modified by TrxG. Concurrently, we discovered that PARG, the enzymatic antagonist of PARP1, co-occupies repressed loci with PcG. Thus, for the first time, our paradigm-shifting data show that PARP1 protein acts as effector machinery, allowing TrxG to maintain chromatin in an activated state.

## 2. Materials and Methods

### 2.1. Histone Peptide Array

The Modified Histone Peptide Array (Active Motif, Carlsbad, CA, USA) was blocked overnight at 4 °C in Tris-buffered saline with Tween 20 (TTBS) buffer (10 mM Tris-HCl, pH 8.3, 0.05% Tween-20, and 150 mM NaCl) containing 5% non-fat milk. Following blocking, the membrane was washed once with TTBS buffer and incubated with 4.0 μg of PARP-1 (Trevigen, Gaithersburg, MD, USA) in PARP binding buffer (10 mM Tris-HCl, pH 8, 140 mM NaCl, 3 mM DTT, and 0.1% Triton X-100) at room temperature for 1 h. The membrane was subsequently washed with TTBS buffer and incubated with a rabbit anti-PARP antibody (ab6079, abcam, Cambridge, UK, 1:500 dilution) in TTBS containing 5% non-fat milk for 1 h at room temperature. Unbound antibodies were removed by washing the membrane three times with TTBS, followed by incubation with horseradish peroxidase-conjugated anti-rabbit antibody (Sigma, St. Louis, MO, USA, 1: 2500 dilution) in TTBS for 1 h at room temperature. The membrane was then treated with ECL developing solution (GE Healthcare, Chicago, IL, USA), and the signal was captured on X-ray film with typical exposure times ranging from 0.5 to 2 min. The signal intensity was quantified using Fiji software (Version 1.54) on an 8-bit grayscale, with background signal subtracted for accurate analysis. This experiment was conducted using two biological replicates.

### 2.2. ChIP-Seq Analysis

ChIP-seq data were analyzed with Galaxy (Version 25.0.3.dev0) [21]. The quality of raw reads was checked using FastQC (Version 0.74), and adapters were removed with fastp (Version 1.0.1) [22]. Trimmed raw reads were aligned to the *Drosophila* genome (dm6) using Bowtie2 (Version 2.5.3) [23]. Unmapped and low-quality reads were discarded (≤20 mapQuality) using BamTools (Version 2.5.2) [24]. Duplicate reads were identified and removed from mapped reads using Picard MarkDuplicates (Version 3.1.1.0). MACS2 (Version 2.2.9.1) [25] was used to call peaks against control (Input or negative control, depending on the dataset) using default settings. For histone marks, broad peaks were called, and narrow peaks for all the other datasets. Peaks were annotated to genomic features with ChIPseeker (Version 1.28.3) [26]. Pairwise correlation of peaks was determined using Intervene (Version 3.5.4) [27]. Normalized coverage of ChIP-seq signals was generated using Deeptools MACS2 bdgcmp (Version 2.2.9.1) [25]. Deeptools multibigwigSummary (Version 3.5.4) and plotCorrelation (Version 3.5.4) [28] were used to determine genome-wide signal correlation using a 1 kb bin size. Deeptools computematrix (Version 3.5.4), Deeptools plotHeatmap (Version 3.5.4), and Deeptools plotProfile (Version 3.5.4) [28] were used with a 50 bp bin size to create enrichment profiles around PARP1 binding region centers (±1.5 kb) in reference mode, or in a 2 kb scale region mode from transcription start site (TSS) to transcription end site (TES) to create enrichment profiles along the genes. Scale region mode also included the 1 kb flanking regions before TSS and after TES. K-means clustering was performed with plotProfile (Version 3.5.4) based on PARP1 distribution.

### 2.3. ATAC-Seq Analysis

Third-instar larvae ATAC-seq data [29] was analyzed as follows. Raw sequencing reads were checked for quality and groomed using Fastp (Version 1.0.1). Quality-checked reads were then mapped to the *Drosophila* genome (dm6) with Bowtie2 (Version 2.5.3), using default parameters and specifying the analysis mode as “very sensitive.” Unmapped and low-quality reads (MAPQ ≤ 30) were removed from the BAM files using BamTools (Version 2.5.2), and mitochondrial reads were excluded. Duplicate mapped reads were identified and removed using Picard MarkDuplicates (Version 3.1.1.0). Peaks were called with MACS2 (Version 2.2.9.1) using a false discovery rate (FDR) threshold of <0.05. Default settings were used with the exception of the “Build Model” parameter, which was disabled (--nomodel), and the shift size was set to −100 (--shift size −100). Finally, the MACS2 bedGraph pileup was converted to bigWig format for visualization using BedGraphToBigWig (Version 2.2.9.1).

### 2.4. Genome-Wide Datasets

We performed a comparative analysis using publicly available ChIP-seq and DamID datasets derived from Drosophila melanogaster across multiple developmental stages and tissues. For each dataset, biological replicates were analyzed independently to confirm reproducibility, after which analyses were conducted using replicate 1. All the datasets were reanalyzed from raw data (see Section 2.2 and Section 2.3 ).

PARP1-YFP: ChIP-seq on wandering third instar larval stage (whole larvae) (GSE217730) [12,30].

PARG-YFP: ChIP-seq on wandering third instar larval stage (whole larvae) (GSE228898) [13].

Su (z)12: ChIP-seq on wandering third instar larval stage (whole larvae) (GSE33546) [31].

E(z): ChIP-seq on wandering third instar larval stage (brain and imaginal disks) (GSE202872) [32].

Jarid2: ChIP-seq on wandering third instar larval stage (whole larvae) (GSE33546) [31].

Pc: DamID on wandering third instar larval stage (salivary glands) (GSE74907) [33].

Psc: ChIP-seq on wandering third instar larval stage (eye imaginal disks) (GSE126985) [34].

Trr: ChIP-seq on 0-5 days adults (whole animals) (GSE89459) [35].

Ash2: ChIP-seq on wandering third instar larval stage (wing imaginal disks) (GSE24115) [36].

Brm, Iswi and Mi-2: ChIP-seq on 15 h-old adult ovaries (GSE168894) [37].

All other ChIP-seq experiments were performed on wandering third instar larval stage (whole animals) and were part of the modENCODE project [38]. These include datasets for H3K4me1 (GSE47282), H3K9me1 (GSE47289), H3K27me1 (GSE49489), H3K27me3 (GSE49490) H3K36me1 (GSE47249), and H4K20me1 (GSE47254).

Time-course gene expression was obtained from reference [39] and measured during the wandering third instar larval stage (Larval puff stage 7–9).

RNA-seq on Trr knockdown data on larval puffstage 7–9 are sourced from [40] (GSE143239).

RNA-seq on Ash2 knockdown data on larval puffstage 7–9 are sourced from [41] (GSE8750).

### 2.5. Statistical Analysis

For Figure 2B–I, statistical analysis involved comparing signal intensities of each histone mark to the corresponding unmodified histone tail fragment. A paired two-tailed *t*-test was applied to assess significance.

For Figure 3C, comparisons were made between each group and the “No Mono” group (negative for all five mono-methylated histone marks: H3K4me1, H3K9me1, H3K27me1, H3K36me1, and H4K20me1). Statistical significance was determined using an unpaired two-tailed *t*-test. Expression values are presented as log2 RPKM. Data were sourced from reference [39].

For Figure 3E, statistical analysis was conducted using the wild-type group as reference. An unpaired two-tailed *t*-test was used to determine significance. Expression values are presented as log2 RPKM. Data were sourced from reference [12].

For Figure 3I, statistical comparisons were made using Cluster 1 as the reference group. Clusters were defined based on k-means clustering of PARP1 binding regions. An unpaired two-tailed *t*-test was used for statistical evaluation.

For Figure 4C, statistical analysis employed an unpaired two-tailed *t*-test with the PRE/TRE group as the reference. PRE (Polycomb Response Elements) and TRE (Trithorax Response Elements) regions were defined as described in reference [42].

For Figure 5B, comparisons were made using the active genes group as reference. An unpaired two-tailed *t*-test was employed. Bivalent genes: Genes marked simultaneously with H3K4me3 and H3K27me3, based on reference [12]. Silent genes: Genes with no detectable expression during the third instar larval puff stage 7–9, as described in reference [39]. Active genes: Genes with expression levels in the 60th percentile or higher during the third instar larval puff stage 7–9.

## 3. Results

### 3.1. Mono-Methylated Histone Tails Bind PARP1 with the Highest Affinity In Vitro

Nucleosomal histones are known to be primary binding partners and regulators of PARP1 in chromatin (Figure 2A) [19,20]. We hypothesize that specific histone modifications will strengthen PARP1 binding affinity. To test this hypothesis, we employed a comprehensive histone peptide array assay, which includes a wide range of modified histone peptides, both individually and in combination [30] (Appendix A). We found that PARP1 binds to the unmodified H3 tail from residues 1 to 19, but not from residues 7 to 26, suggesting that residues 1 to 7 are crucial for PARP1 binding (Figure 2B). PARP1 showed no binding to unmodified H4 or H2B histone tails; however, it did bind to the unmodified H2A tail from residues 1 to 19, consistent with our previous findings [18,19,20].

We observed that PARP1 binding is strongly affected by the presence of mono-methylated residues. Specifically, PARP1 strongly binds to H4K20me1 (Figure 2C,D), and H3K4, H3K9, H3K27, and H3K36 mono-methylation dramatically increases PARP1 binding to Histone 3 (Figure 2E–G). Remarkably, PARP1 binding to mono-methylated H3 was further boosted by phosphorylated threonine 3 (T3P) but suppressed by S10P/T11P and S28P. PARP1 did not bind to the modified H2B tail under any tested condition (Figure 2H). PARP1 affinity to the H2A tail decreased in the presence of phosphorylated serine 1 (S1P) or acetylated lysine 5 (K5ac), while the simultaneous presence of K5ac and K13ac significantly increased PARP1 binding (Figure 2I). In summary, PARP1 exhibited the highest affinity for mono-methylated histones, while other modifications generally suppressed these interactions (Figure 2J). These findings suggest that the presence of mono-methylated histone marks could significantly influence PARP1 binding patterns to chromatin in vivo.

### 3.2. Mono-Methylated Histones Control PARP1 Binding at PARP1-Controlled Loci Genome-Wide

To validate our cell-free system findings in vivo, we investigated the genome-wide landscape of PARP1 binding. We observed that PARP1 exclusively binds to loci positive for any of the mono-methylated histone marks, including H3K4me1, H3K9me1, H3K27me1, H3K36me1, or H4K20me1 (Figure 3A, Appendix A). During gene activation, PARP1 rapidly spreads along the gene body to facilitate transcription [17,18]. To elucidate the molecular mechanisms controlling this spreading, we examined various combinations of the five noted mono-methylated histone marks. We found that PARP1 spreads along the gene body of loci positive for at least two of the marks among H3K4me1, H3K9me1, and H4K20me1 (Figure 3B, Appendix A). These loci exhibit high transcriptional activity, present a more open chromatin state, and are severely transcriptionally downregulated in the absence of PARP1, suggesting that PARP1 is critical for their transcriptional activation (Figure 3C–E, Appendix A).

We hypothesized that mono-methylated histone marks guide the spreading of PARP1 along the gene body to activate and maintain gene expression. Consequently, we expected mono-methylated marks to extend deeper into the gene body of highly active loci. However, our analysis revealed that mono-methylated histone marks are not uniformly distributed along PARP1 binding sites. Specifically, H3K9me1 and H4K20me1 peak at approximately 150 bp downstream of PARP1 binding sites, while H3K36me1, H3K4me1, and H3K27me1 peak at 350 bp, 750 bp, and 1450 bp downstream, respectively (Figure 3F). Notwithstanding these results, a deeper distribution of H3K4me1, H3K9me1, and H4K20me1 was found along the gene bodies of genes marked by this combination as PARP1 spreads (Figure 3G, Appendix A). To further confirm the relationship between PARP1 and mono-methylated histone marks, we performed k-means clustering on PARP1 binding regions. We identified three distinct distribution patterns of PARP1 (Figure 3H, Appendix A). Cluster 1 presents the highest spreading of PARP1 along the gene body, and this correlates with genes having the highest expression levels during third instar larvae, thus confirming that PARP1 spreads along active loci (Figure 3I). Moreover, all five mono-methylated histone marks noted above show deeper spreading along Cluster 1 gene bodies compared to the other clusters (Figure 3J,K) (Appendix A). H3K9me1 and H4K20me1 peaked at + 180 base pairs downstream of the TSS in both highly and moderately expressed genes (Clusters 1 and 2) but only showed strong occupancy in the gene bodies of highly expressed genes (Figure 3J, Appendix A). Conversely, H3K4me1 and H3K36me1 peaked around the + 680 bp and + 310 bp regions downstream of the TSS in Cluster 2 genes, respectively, while peaking at 1.3 kb in Cluster 1 genes (Figure 3K). Additionally, H3K27me1 was enriched at the promoter region of low-expressed genes (Cluster 3) but showed strong enrichment at the gene body of highly expressed genes, peaking around + 1.7 kb downstream of the TSS (Figure 3K). These data demonstrate that mono-methylated histone marks control PARP1 binding and spreading along gene bodies, a process essential for the activation and maintenance of PARP1-mediated transcriptional loci (Figure 3L) (Appendix A). Next, we investigated the complexes responsible for activating gene expression through the deposition of specific histone modifications.

### 3.3. PARP1 Controls the Expression of Developmental Genes Controlled by TrxG

Mono-methylated histone marks H3K4me1, H3K9me1, H3K27me1, H3K36me1, and H4K20me1 are typically associated with active chromatin [43]. The Trithorax Group (TrxG) comprises complexes that label chromatin as active through the deposition of specific histone modifications marks [3]. In *Drosophila*, the Trithorax Group (TrxG) comprises Trithorax (Trx), dSET1, and Trithorax-related (Trr) complexes, all sharing Ash2 as an essential cofactor [44,45,46]. To investigate the association between mono-methylated histone marks H3K4me1, H3K9me1, H3K27me1, H3K36me1, and H4K20me1 and TrxG, we analyzed their occupancy on TrxG-positive and TrxG-negative loci. Our results revealed a significant enrichment of all mono-methylated histone marks, except for H3K27me1, at TrxG-positive loci, indicating a strong association with TrxG (Figure 4A) (Appendix A). Polycomb Group (PcG) complexes, another family of chromatin regulators, have been traditionally associated with chromatin compaction, but as noted above, recent studies suggest a role in activation as well [3,4]. In *Drosophila*, Polycomb Group (PcG) complexes include Polycomb Repressive Complexes (PRC) 1 and 2, which are responsible for H2AK118 ubiquitination and H3K27 methylation, respectively [47,48], along with Pho Repressive Complex (Pho-RC) [49].

We further investigated the interaction between PARP1 and both PcG and TrxG. In *Drosophila*, PcG and TrxG are recruited via cis-regulatory regions called Polycomb response elements (PRE) and Trithorax response elements (TRE), respectively. While our results show that PARP1 binds exclusively to loci containing TRE, we observed a stronger affinity for loci bearing both PRE and TRE (Figure 4B,C). Further investigation revealed that PARP1, PcG, and TrxG interact such that PARP1 binds exclusively to loci positive for either PcG or TrxG, but with a pronounced preference for TrxG (Figure 4D,E, Appendix A). These findings indicate that PARP1 predominantly binds to loci controlled by TrxG with a lesser degree of association with PcG-controlled loci.

Since PARP1 binds to loci bearing TrxG, we explored the potential cooperative role of these proteins in transcriptional regulation. Previously, we reported that PARP1 influences the expression of developmental genes at the end of the third instar larval stage (puff stages 7–9) [12,13]. Here, we found that PARP1 and TrxG promote the expression of the same set of genes involved in developmental processes, such as cell differentiation and morphogenesis (Figure 4F,G, Appendix A). These results indicate that PARP1 controls the expression of developmental TrxG-dependent loci (Figure 4H).

### 3.4. PARG Controls the Expression of PcG-Controlled Loci

In contrast to the role of PARP1 in transcriptional activation, Poly (ADP-ribose) glycohydrolase (PARG), which is responsible for pADPr catabolism, plays a role in transcriptional repression [13,50]. We found that PARG preferentially binds to the loci bearing H3K27me3 (Figure 5A, Appendix A), we then hypothesized that PARG binds to loci controlled by PcG proteins. We found that PARG binds exclusively to loci positive for either PcG or TrxG proteins with a strong preference for PcG proteins (Figure 5B, Appendix A). This association is further supported by the enrichment of PARG and the PcG component Polycomb (Pc) at repressed genes, in contrast to the enrichment of PARP1 and TrxG components, including Brahma (Brm), Iswi, and Mi-2, at active genes (Figure 5C–L, Appendix A).

Notably, PcG, TrxG, and Swi/SNF components are all enriched at bivalent genes, which harbor both the active histone modification H3K4me3 and the repressive mark H3K27me3 (Figure 5E–L, Appendix A). PARG also demonstrates a preference for bivalent genes, whereas PARP1 shows similar levels of occupancy at both active and bivalent genes (Figure 5D,E, Appendix A).

These data support a model in which PARG represses loci controlled by Pc itself, acting antagonistically to PARP1, which promotes the activation of TrxG-regulated loci (Figure 5M). Furthermore, our data suggests a collaborative interplay among PARP1, PARG, TrxG, and PcG proteins in regulating bivalent genes, highlighting their roles in balancing activation and repression at these loci.

## 4. Discussion

Twenty-five years ago, David Allis proposed the revolutionary idea that histone posttranslational modifications form a “histone code” established by “writer” complexes to mark specific loci, facilitating the recruitment of “reader” complexes that regulate chromatin conformation [8]. Regions destined for chromatin compaction are marked differently from those destined for chromatin opening. Consequently, over the past two decades, research has focused on identifying the readers recruited by this histone code to influence chromatin structure. Recent advances have identified such readers involved in chromatin compaction, providing a clearer understanding of how this process occurs. For instance, the bromo adjacent homology domain-containing protein 1 (BAHD1) is a reader of the H3K27me3 mark. As such, it binds to H3K27me3-positive loci and acts as a scaffold protein, promoting heterochromatin formation [51]. Furthermore, recent studies suggest that Polycomb Group proteins (PcG), which write the H3K27me3 mark, also act as readers of H3K27me3, also forming a scaffold that recruits proteins, such as bromodomain-containing protein 4 (Brd4), to compact chromatin [10]. Collectively, these studies have furthered our understanding of transcriptional silencing via chromatin compaction.

However, progress has been less promising in elucidating the mechanisms underlying transcriptional activation via chromatin opening. While SWI/SNF complexes can slide nucleosomes along DNA to facilitate local opening, they cannot account for the extensive chromatin opening required during developmental puffs. In light of the absence of effective readers that loosen chromatin, some researchers have proposed that histone posttranslational modifications themselves might directly affect chromatin remodeling. For example, histone acetylation could decrease the affinity of histones for DNA [52]. Over time, the histone code hypothesis has been ignored. Our results strongly support Allis’s original histone code hypothesis by identifying Poly (ADP-ribose) Polymerase 1 (PARP1) and Poly (ADP-ribose) Glycohydrolase (PARG) as major “reader–effector” proteins: PARP1 acting downstream of TrxG and PARG acting downstream of PcG. We demonstrate that PARP1 is recruited to TrxG-deposited mono-methylated histone marks and spreads along gene bodies to loosen chromatin during transcription, while PARG preferentially associates with PcG-bound loci, reinforcing repressive chromatin states.

Since their discovery, Polycomb complexes have been extensively studied for their involvement in transcriptional repression. However, their role in transcriptional activation in conjunction with Trithorax complexes has garnered less attention. Notably, Polycomb complexes were previously found to be enriched at the promoter region of active genes [34,53,54,55]. In the present study, however, we specifically found that the chromatin-associated protein PARP1 binds extensively to both Polycomb and Trithorax response elements (PRE/TRE), colocalizing with several members of Polycomb and Trithorax complexes at the promoters of active genes. Additionally, in a cell-free system, we demonstrated that PARP1 actively binds to mono-methylated histones, particularly H3K4me1 and H3K27me1, deposited by Trithorax complexes and PRC2, respectively [30]. These two marks are strongly correlated with the presence of PARP1 (Figure 3B). Collectively, these results support our stated hypothesis that PARP1 protein, along with TrxG, acts as effector machinery to maintain chromatin in the activated state and suggest that PARP1 plays a significant role in regulating target genes shared with Polycomb and Trithorax complexes.

Interestingly, previous studies aiming to identify Polycomb complex interactors have neglected to recognize PARP1 as an interacting player [56,57,58]. Our data indicates that PARP1 extensively binds to PRE/TRE, suggesting that Polycomb/Trithorax complexes and PARP1 occupy the same loci, albeit in a temporally distributed manner along the landscape of PARP1-bound gene bodies. Based on these observations, we propose a de novo model to explain the interaction between Polycomb/Trithorax complexes and PARP1, while noting that several mechanisms have already been proposed to maintain the repressive state, including the recruitment of bromodomain-containing protein 4 (Brd4) through its interaction with Embryonic Ectoderm Development (EED), a member of PRC2 [10]. Here, we reasoned that transcriptional repression by Polycomb complexes begins with the recruitment of PcG to PRE/TRE regions, leading to the trimethylation of H3K27 by PRC2 and the ubiquitination of H2AK118 by PRC1, which, in turn, results in chromatin compaction. Conversely, transcriptional activation begins with the recruitment of Polycomb and Trithorax complexes to PRE/TRE regions, resulting in the deposition of H3K4me3 by Trithorax complexes and the mono-methylation of histones, such as H3K4me1 by Trithorax complexes and H3K27me1 by PRC2. Then, the presence of mono-methylated histones facilitates the recruitment of PARP1, which, upon activation, undergoes automodification and loosens chromatin, thereby maintaining the active state.

Not all Polycomb complex subunits colocalize with PARP1. For example, Polycomb (Pc), a PRC1 subunit that detects H3K27me3 to recruit PRC1, is completely depleted from PARP1 binding regions, as we demonstrated in our results. Interestingly, the presence of H3K27me3 is not necessary for the recruitment of Polycomb complexes to PRE [59], indicating that the presence of Polycomb complexes at PARP1 binding regions may be atypical.

Finally, we found that mono-methylated histone marks are enriched at PARP1 binding regions and are spatially distributed along the landscape of PARP1-bound gene bodies. This observation aligns with our cell-free system results, which demonstrate that PARP1 binds to histone tails only in the presence of monomethylated histone marks. Upon gene activation, PARP1 is activated and spreads quickly along the gene body [17,18]. Since PARP1 binds to mono-methylated histones [30], this unique distribution could act as a guide for the spreading of PARP1 during transcription activation. This discovery highlights the pivotal role of monomethylated histone marks in orchestrating PARP1 recruitment and spreading, offering new insights into the mechanisms of transcriptional regulation in eukaryotes during development.

## 5. Conclusions

The histone code hypothesis proposed that specific histone modifications recruit effector proteins to shape chromatin states. While this concept has guided decades of research on transcriptional repression, the identity of effectors driving chromatin opening has remained elusive. Here, we identified PARP1 as a key reader of mono-methylated histone marks deposited by Trithorax and Polycomb complexes. We showed that PARP1 binds to PRE/TREs and spreads across gene bodies to loosen chromatin and promote transcription. Our findings revive and extend the histone code model by demonstrating how mono-methylation recruits PARP1 to drive gene activation, revealing a new layer of regulation in the establishment and maintenance of developmental transcriptional programs.

## Figures and Tables

**Figure 1 biomolecules-15-01314-f001:**
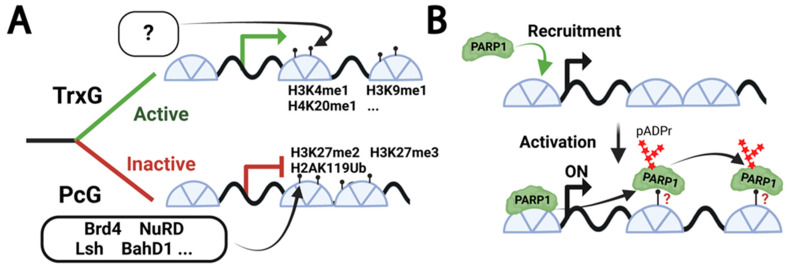
**Chromatin Regulation by TrxG/PcG Complexes and PARP1.** (**A**) Scheme illustrating the mechanisms of transcriptional activation and silencing by Trithorax (TrxG) and Polycomb (PcG) complexes, respectively. TrxG action results in the accumulation of active histone marks, such as the mono-methylation of H3K4, H3K9, and H4K20 residues. The effectors that are reading TrxG-deposited marks to activate and maintain active specific loci remain unclear. Conversely, PcG action leads to the accumulation of repressive histone marks, including H3K27me2/3 and H2AK119 ubiquitination. PcG effectors such as Brd4, NuRD complex, Lsh, BahD1, and more read PcG-deposited marks to repress and maintain close specific loci. Green and red lines indicate active and repressed genes, respectively. (**B**) This schematic illustrates the current model for PARP1 recruitment and activation. PARP1 binds to nucleosomes. Upon activation, PARP1 spreads along the gene body to facilitate chromatin loosening. While the exact mechanism of PARP1 spreading remains unclear, the presence of specific histone marks along the gene body might contribute to this process. Curved green arrow: initial PARP1 recruitment; curved black arrows: PARP1 spreading along the gene body.

**Figure 2 biomolecules-15-01314-f002:**
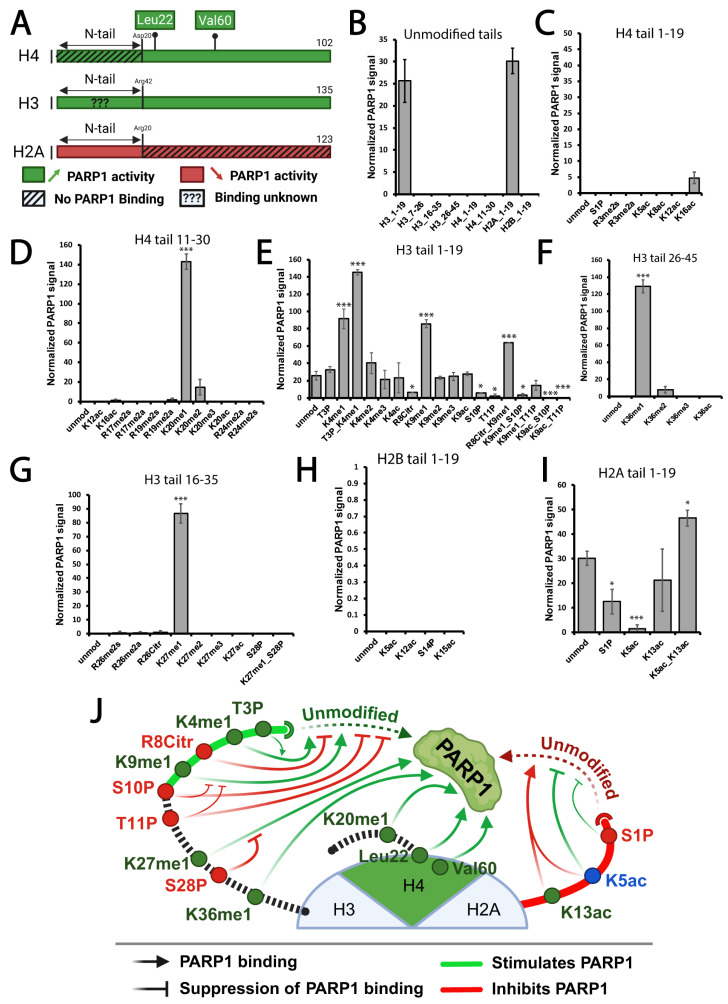
**Histone modification code controls PARP1 protein binding.** (**A**) Schematic representation of PARP1 binding patterns to histones. PARP1 binds to the core of histone H4, but not its tails, with leucine 22 and valine 60 being essential for this binding. PARP1 binds to the core of histone H3 and to the tail of H2A. (**B**) Signal intensity on unmodified 19mer histone tail peptides after incubation with PARP1 protein. Only the 1–19 fragments of H3 and H2A exhibit signal intensities higher than the background. (**C**–**I**) Signal intensity on modified 19mer histone tails after incubation with PARP1 protein: H4 1–19 (**C**), H4 11–30 (**D**), H3 1–19 (**E**), H3 26–45 (**F**), H3 16–35 (**G**), H2B 1–19 (**H**) or H2A 1–19 (**I**). ***: *p*-value < 0.01, *: *p*-value < 0.05 (See Section 2 for detail). Error bars represent the standard error of the mean (S.E.M.). (**J**) Schematic Representation of Histone Marks Influencing PARP1 Binding. Light green and light red arrows represent the promotion and inhibition of PARP1 binding, respectively. Size of the arrows indicates the extent to which PARP1 binding is influenced. Histone marks that facilitate or suppress PARP1 binding are indicated in green and red, respectively. Histone marks performing both are indicated in blue. Dashed dark green and dark red arrows depict PARP1 binding to the unmodified tails of H2A and the beginning of H3 tails, respectively. Black dashed lines illustrate the lack of PARP1 binding to unmodified H4 tails and the end of H3 tails. PARP1 binding to the H2A tail inhibits PARP1 activity. In this context, K5ac suppresses PARP1 binding to H2A tails, thereby positively regulating PARP1 activity, as illustrated by a green inhibitory arrow. Conversely, the simultaneous presence of K5ac and K13ac boosts PARP1 affinity to H2A tails, resulting in the negative regulation of PARP1 activity, as illustrated by a red arrow.

**Figure 3 biomolecules-15-01314-f003:**
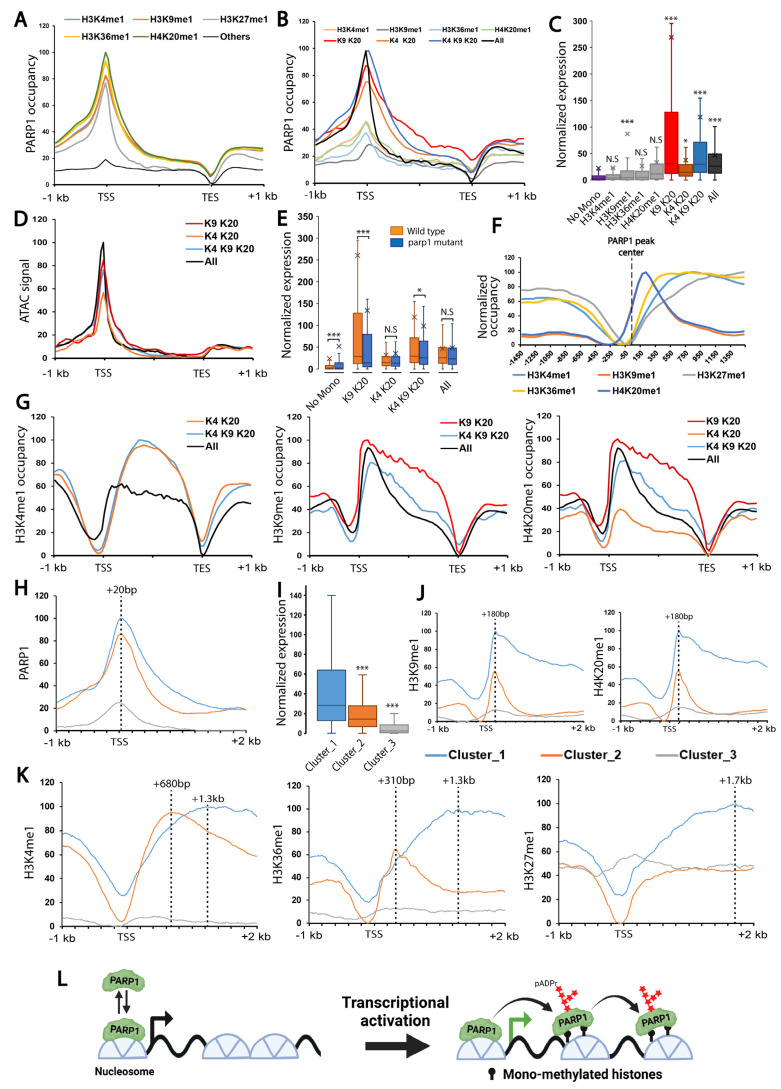
**Mono-Methylated Histone Marks control PARP1 binding at PARP1-controlled loci genome-wide.** (**A**) PARP1 binds exclusively to loci bearing mono-methylated histone marks (H3K4me1, H3K9me1, H3K27me1, H3K36me1, or H4K20me1). Curve representing PARP1 distribution in loci lacking these mono-methylated histone marks is shown in black. Data are normalized from 0 (minimal signal) to 100 (maximal signal). A locus was considered positive for a given mono-methylated histone mark if a significant enrichment peak was detected within the region spanning from −500 bp upstream of the transcription start site (TSS) to the transcription end site (TES). (**B**) PARP1 distribution across loci marked with combinations of mono-methylated histones: H3K9me1 + H4K20me1 (red), H3K4me1 + H4K20me1 (orange), and H3K4me1 + H3K9me1 + H4K20me1 (blue). At loci carrying all five marks (black), PARP1 is restricted to promoter regions. Binding is weak at loci carrying only a single mono-methyl mark (light orange, gray, light blue, light green). (**C**) Loci bearing a combination of histone marks between H3K4me1, H3K9me1, and H4K20me1 are highly transcriptionally active compared to other groups. Box plots show data distribution (25th–75th percentile), whiskers represent the range, and the *y*-axis indicates log2 RPKM (reads per kilobase per million mapped reads). (**D**) ATAC-seq signal in third instar larvae, revealing greater chromatin accessibility of genes positive for H3K9me1 and H4K20me1 (red), H3K4me1 and H4K20me1 (orange), and H3K4me1, H3K9me1, and H4K20me1 (blue). (**E**) The expression of the genes bearing a combination of H3K4me1, H3K9me1, and H4K20me1 is parp1-dependent. Date sourced from reference [12]. (**F**) Profiles of monomethylated histone marks around PARP1 binding sites. PARP1 peaks are oriented based on the strand of the gene in which the PARP1 peak is located (plus or reverse strand). The center of the PARP1 peaks is marked with a dashed line. Curves represent the distribution of monomethylated histone marks within a 3 kb region flanking the PARP1 peak center (−1.5 kb to + 1.5 kb). The *x*-axis denotes the distance from the PARP1 peak center in base pairs, while the *y*-axis represents the signal intensity of the histone marks normalized from 0 (minimal signal) to 100 (maximal signal). (**G**) Distribution of H3K4me1 (left), H3K9me1 (middle), and H4K20me1 (right) in the presence or absence of other monomethylated histone marks. (**H**) Distribution of PARP1 across the gene bodies of genes in Cluster 1 (blue), Cluster 2 (orange), and Cluster 3 (gray). (**I**) Gene expression levels differ significantly among clusters in third instar larvae, with Cluster 1 showing the highest expression, Cluster 2 intermediate, and Cluster 3 the lowest. Expression data are from [39]. (**J**,**K**) Distribution of histone marks across gene bodies in clusters: (**J**) H3K9me1 and H4K20me1; (**K**) H3K4me1, H3K36me1, and H3K27me1. (**L**) Schematic depicting the interaction between mono-methylated histone marks and PARP1 during transcriptional activation. Prior to gene activation, PARP1 is localized at the promoter region. Upon transcriptional activation, PARP1 rapidly spreads along the gene body guided by mono-methylated histone marks. Straight black arrows: initial PARP1 recruitment; Curved black arrows: PARP1 spreading along the gene body. Statistical significance: *** *p* < 0.01, * *p* < 0.05, N.S. (not significant). See Materials and Methods for further details.

**Figure 4 biomolecules-15-01314-f004:**
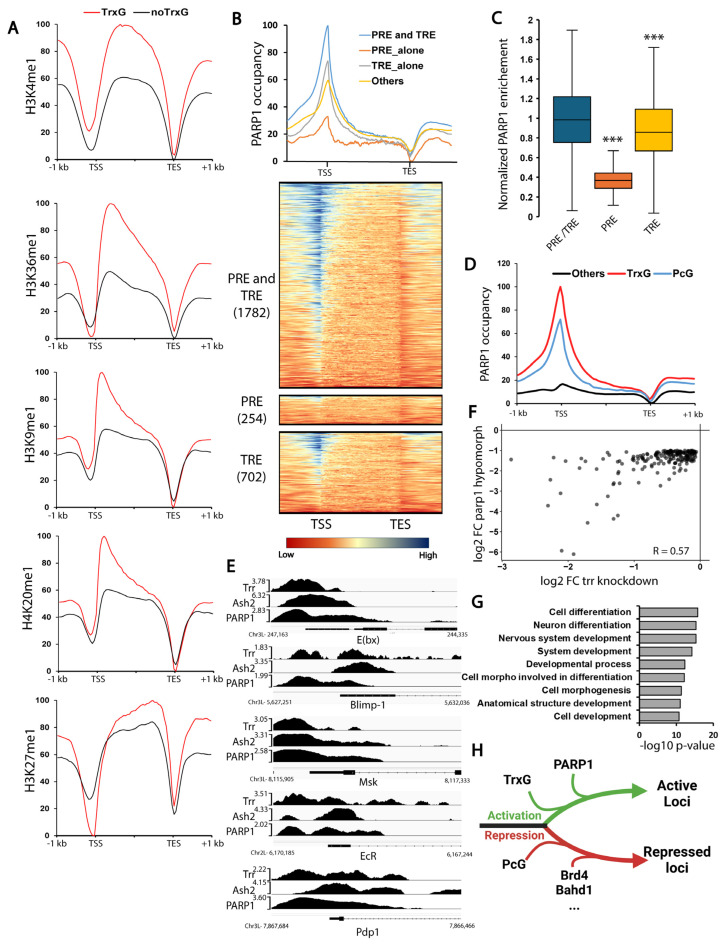
**PARP1 controls the expression of TrxG-dependent loci.** (**A**) Mono-methylated histone marks, with the exception of H3K27me1, present a stronger enrichment at loci positive to TrxG. TrxG-positive loci are defined by the presence of at least one of the following Trr, Ash2, Brm, or Iswi. (**B**) PARP1 binds selectively to loci containing Trithorax response elements (TREs) as well as loci harboring both Polycomb (PREs) and Trithorax (TREs) response elements, as defined in reference [42]. PARP1 occupancy is shown as a color gradient from red (lowest) to blue (highest). (**C**) Box plot representing PARP1 enrichment to PRE/TRE, and TRE regions. Enrichment was calculated by comparing the number of PARP1-GFP reads between a line expressing PARG-GFP and a control line lacking GFP. Data were normalized to the average enrichment observed for PRE/TRE regions. Statistical significance determined by an unpaired two-tailed *t*-test. ***: *p*-value < 0.01. (**D**) PARP1 binding is restricted to loci enriched for TrxG or PcG subunits. TrxG-positive loci are defined by the presence of Trr and/or Ash2, whereas PcG-positive loci are defined by the presence of one or more of the following: Psc, Ph, Pc, E (z), Su (z)12, Jarid2, Pho, or Sfmbt. (**E**) Integrated Genome Viewer (IGV) tracks illustrating the distribution of PARP1 and the TrxG components Ash2 and Trr along the promoter region of five representative loci. The Trr dataset comes from adult flies and not third instar larvae. (**F**) Scatter plot of transcriptional expression profiles for 194 genes downregulated in both trr knockdown (RNAi) and parp1 hypomorphic conditions during the wandering third instar larval stage (puff stage 7–9). Data are from references [12,40]. The correlation coefficient for changes in gene expression is 0.57, indicating a positive correlation. Correlation was calculated by comparing log2 fold changes in gene expression between wild-type and *trr* knockdown animals and between wild-type and *parp1* knockdown animals. Fold change was calculated from RPKM values in wild-type versus knockdown conditions. (**G**) Gene Ontology (GO) analysis of the 194 co-downregulated genes. A total of 64 enriched biological process terms were identified; all related to differentiation and morphogenesis programs. The figure displays the top nine most significant terms with their −log10 *p*-values. (**H**) Schematic summary of Figure 4: PARP1 controls the expression of TrxG-dependent loci. Green and red lines indicate active and repressed genes, respectively. (See Section 2 for detailed protocols).

**Figure 5 biomolecules-15-01314-f005:**
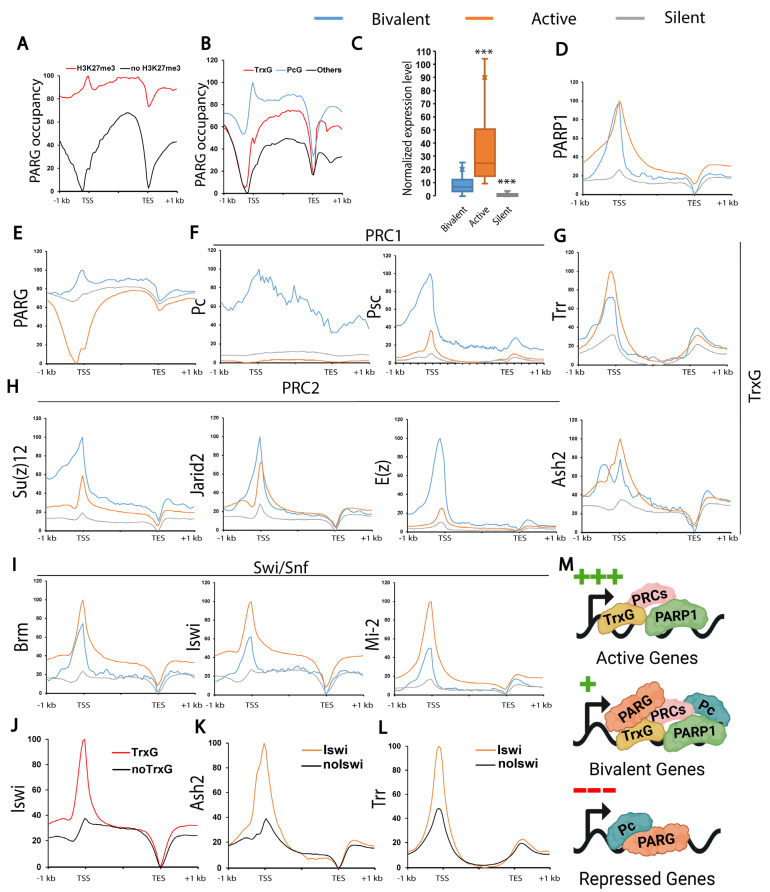
**PARP1 and TrxG co-occupy active loci, while PARG and PcG co-occupy repressed loci**. (**A**) PARG preferentially binds to loci bearing H3K27me3. (**B**) PARG preferentially binds to PcG or TrxG-positive loci. (**C**) This plot displays gene expression levels for bivalent, active, and silent genes at the end of the third instar larval stage (puffstage 7–9). ***: *p*-value < 0.01. (**D**–**I**) These panels illustrate the distribution of PARP1 (**D**), PARG (**E**), PRC1 components Pc and Psc (**F**), Trithorax components Trr and Ash2 (**G**), PRC2 components Su (z)12, Jarid2, and E (z) (**H**), Brm, Iswi, and Mi-2 (**I**), across the gene bodies of bivalent genes (blue), active genes (orange), and silent genes (gray). (**J**) Distribution of Iswi along the loci positive (red) or negative (black) for TrxG. TrxG-positive loci are defined by the presence of Trr or Ash2. (**K**,**L**) Distribution of Trr (**K**) or Ash2 (**L**) along the loci positive (red) or negative (black) for Iswi. (**M**) Model summarizing Figure 5: Active genes (green +++) exhibit the presence of TrxG and PcG, along with PARP1, while PARG and Polycomb proteins are absent. Bivalent genes (green +) display the presence of TrxG and PcG, along with PARP1, and feature PARG and Polycomb. Silent genes (red -) are characterized by the presence of only Polycomb protein and PARG. Trr and Brm/Iswi/Mi-2 datasets come from adult flies and not third instar larvae (See Section 2 for detailed protocols).

## Data Availability

No new data were created or analyzed in this study. Data sharing is not applicable to this article.

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
