# Peer review of "PARP1 and PARG Are the Draft Horses for Polycomb-Trithorax Chromatin Regulator Machinery"

_biomolecules, 2025, doi:10.3390/biom15091314_

Round 1

Reviewer 1 Report

Comments and Suggestions for Authors

The article “PARP1 is the draft horse for Polycomb-Trithorax chromatin regulator machinery” by G. Bordet and A.Tulin is devoted to the study of the PARP1 role in maintaining chromatin in an activated loosen state and the role of PARG in transcriptional repression.

The key notes:

  1. The article analyzes a large amount of data, but it is completely unclear how it was obtained. In particular, the sections “ChIP-seq analysis” and “ATAC-seq analysis” describe the bioinformatics analysis, but there is no description of the acquisition of primary data or reference to their source. Section "3.2. Mono-methylated histones control PARP1 binding at PARP1-controlled loci genome-wide" should describe the course of the experiment. How was loci positive for the mono-methylated histone marks determined? How was the gene expression level determined (for clustering)? Authors should describe how data on TrxG or PcG-positive loci were obtained. TrxG and PcG are protein complexes, are we talking about a specific component or the entire complexes? Since the origin of some of the data is not described, the reader cannot draw his own conclusions and is forced to rely entirely on the interpretation of the data proposed by the authors.
  2. The title "PARP1 is the draft horse for Polycomb-Trithorax chromatin regulator machinery" does not fully reflect the content of the article. The article shows that PARP1 controls the expression of TrxG -controlled loci, and PARG controls the expression of PcG-controlled loci. Perhaps it is worth indicating in the title that the PARP1 and PARG antagonist enzymes control opposite processes in the regulation of Polycomb-Trithorax chromatin machinery? In general, it seems to me that PARG is undeservedly ignored in the title, in the discussion, and in the conclusion.

Minor points:

  1. In the Abstract, do “repressive or active histone code marks” mean “suppressive or activating marks”?
  2. Why was the third instar larval stage chosen?
  3. Fig.2: How is fig 2A different from fig 2B? Between 2G and 2D, where do the color designations belong? In fig 2G, in the middle (for H3K9me1), there are no color designations at all.
  4. Fig.3: How is 3B different from 3D?

Author Response

Reviewer 1: The article “PARP1 is the draft horse for Polycomb-Trithorax chromatin regulator machinery” by G. Bordet and A.Tulin is devoted to the study of the PARP1 role in maintaining chromatin in an activated loosen state and the role of PARG in transcriptional repression.

The key notes:

The article analyzes a large amount of data, but it is completely unclear how it was obtained. In particular, the sections “ChIP-seq analysis” and “ATAC-seq analysis” describe the bioinformatics analysis, but there is no description of the acquisition of primary data or reference to their source. Section "3.2. Mono-methylated histones control PARP1 binding at PARP1-controlled loci genome-wide" should describe the course of the experiment. How was loci positive for the mono-methylated histone marks determined? How was the gene expression level determined (for clustering)? Authors should describe how data on TrxG or PcG-positive loci were obtained. TrxG and PcG are protein complexes, are we talking about a specific component or the entire complexes? Since the origin of some of the data is not described, the reader cannot draw his own conclusions and is forced to rely entirely on the interpretation of the data proposed by the authors.

Response: We thank Reviewer 1 for these insightful comments. We have extensively revised the manuscript to include detailed descriptions of the data sources and experimental information. Specifically, we now provide (i) a clear description of the origin of all primary datasets in the Materials and Methods (Genome-wide datasets section), (ii) an explanation of how loci positive for mono-methylated histone marks were defined, (iii) details on how gene expression levels were obtained for the different cluster, and (iv) clarification regarding the identification of TrxG- and PcG-positive loci, specifying the individual subunits analyzed. These revisions ensure that the datasets, their origin, and the analytical framework are transparent and reproducible.

R1: The title "PARP1 is the draft horse for Polycomb-Trithorax chromatin regulator machinery" does not fully reflect the content of the article. The article shows that PARP1 controls the expression of TrxG -controlled loci, and PARG controls the expression of PcG-controlled loci. Perhaps it is worth indicating in the title that the PARP1 and PARG antagonist enzymes control opposite processes in the regulation of Polycomb-Trithorax chromatin machinery? In general, it seems to me that PARG is undeservedly ignored in the title, in the discussion, and in the conclusion.

Response: We have revised the title, abstract, results, and discussion to better reflect the dual roles of PARP1 and PARG as effectors of the Trithorax and Polycomb machineries, respectively. These revisions ensure that the contribution of PARG is explicitly acknowledged and integrated throughout the manuscript.

Minor points:

R1: In the Abstract, do “repressive or active histone code marks” mean “suppressive or activating marks”?

Response: The terms repressive and active histone marks are widely used shorthand to indicate histone modifications associated with transcriptional silencing or activation, respectively.

R1: Why was the third instar larval stage chosen?

Response: The wandering third instar larval stage (puff stage 7–9) represents a period of extensive transcriptional reprogramming. At this stage, metabolic and larval developmental genes are repressed (controlled by PARG), while pupal developmental genes are activated (driven by PARP1). Importantly, PARP1 activity reaches its peak during this transition, making this developmental window ideally suited to study PARP1-dependent transcriptional activation.

R1: Fig.2: How is fig 2A different from fig 2B? Between 2G and 2D, where do the color designations belong? In fig 2G, in the middle (for H3K9me1), there are no color designations at all.

Response: Figure 2A illustrates PARP1 distribution at loci carrying any mono-methylated histone marks, whereas Figure 2B focuses on PARP1 distribution at loci with specific combinations of these marks. We have revised the figure legend to better highlight this distinction. In addition, we corrected the color designations in Figures 2D and 2G and updated Figure 2G to include the missing label.

R1: Fig.3: How is 3B different from 3D?

Response: Figure 3B presents PARP1 distribution along loci marked by Polycomb and/or Trithorax response elements (PRE/TRE), whereas Figure 3D shows PARP1 distribution along loci enriched for Polycomb and/or Trithorax complex subunits. We have revised the figure legend to make this distinction clearer. We thank Reviewer 1 for their careful reading and valuable comments.

Reviewer 2 Report

Comments and Suggestions for Authors

Although the hypotheses and related discussions are indeed appealing, the paper relies on a single experiment conducted by the group—the peptide array—for which the authors only show the analyzed data. I would prefer to also see the raw film exposure. The re-analysis of ATAC-seq and ChIP-seq data from other sources suffers from the fact that several experiments with different coverage levels are being compared, making it difficult to draw meaningful comparisons. My concern increases upon noticing that different embryonic/adult stages of Drosophila are being compared.

The sentence in the abstract, "PARP1 binds TrxG-generated histone marks with high affinity in vitro, completely colocalizing with them genome-wide, and controls the expression of all TrxG-dependent loci", is not cautious, as the authors have not demonstrated that PARP1 binds TrxG-generated histone marks, nor have they proven that PARP1 controls the expression of TrxG-dependent genes. Only expression correlation analyses were performed, with no evidence of causality.

Had the authors been able to perform in vivo experiments in Drosophila, the quality of the paper would have increased exponentially. However, if that is not possible, I suggest the following:

  • Fig. 1 D–H: I’d like to see the original data (the scan of the film), at least as a supplementary file.
  • Clearly indicate in the manuscript which NGS experiments are being re-analyzed, specifying the developmental stage of Drosophila in each case.
  • The differences in binding and signal in ChIP-seq shown in Figures 2a, b, g, h, j, k; 3a, b, d; and 4a, b, e, f, g, h, i, j should be quantified using appropriate tools such as DiffBind. Qualitative plots are not sufficient, and the figures need to be completely restructured.
  • It should be clearly stated that the findings are correlational and not causal, with a much more cautious tone, and that the conclusions are valid only in Drosophila.
  • The tracks in Fig. 3e do not appear to be of good quality; a reference to the genomic location and signal intensity should be included.
  • For all analyzed NGS experiments, the replicates used must be specified

Author Response

Reviewer 2: Although the hypotheses and related discussions are indeed appealing, the paper relies on a single experiment conducted by the group—the peptide array—for which the authors only show the analyzed data. I would prefer to also see the raw film exposure. The re-analysis of ATAC-seq and ChIP-seq data from other sources suffers from the fact that several experiments with different coverage levels are being compared, making it difficult to draw meaningful comparisons. My concern increases upon noticing that different embryonic/adult stages of Drosophila are being compared.

Response: We thank Reviewer 2 for their constructive comments. With regard to dataset selection, all datasets analyzed in this study were derived from the wandering third instar larval stage, with the exception of Trr and Brm/Iswi/Mi-2, for which only adult datasets are available. Although this introduces a stage-specific distinction, the distribution of Trr in adult tissues closely mirrors that of Ash2 in third instar larvae, showing comparable enrichment patterns at bivalent, active, and silent genes. Importantly, functional evidence supports this parallel: genetic knockdown of Trr in third instar larvae leads to downregulation of developmental genes, many of which are also downregulated upon parp1 knockdown. This result is further reinforced by our new analysis of Ash2 mutants in third instar larvae, which reveals an even stronger correlation with parp1 knockdown (see new Supplemental Figure S8). Together, these results provide convergent lines of evidence that support a robust functional association between PARP1 and TrxG proteins, despite the unavoidable stage-specific differences in available datasets.

R2: The sentence in the abstract, "PARP1 binds TrxG-generated histone marks with high affinity in vitro, completely colocalizing with them genome-wide, and controls the expression of all TrxG-dependent loci", is not cautious, as the authors have not demonstrated that PARP1 binds TrxG-generated histone marks, nor have they proven that PARP1 controls the expression of TrxG-dependent genes. Only expression correlation analyses were performed, with no evidence of causality.

Response: We have revised the abstract sentence to read: “We found that PARP1 binds TrxG-generated histone marks with high affinity in vitro, completely colocalizing with them genome-wide, and controls the expression of loci modified by TrxG.” Importantly, this conclusion is supported by multiple independent lines of evidence. First, genetic knockdown of Trr or Ash2, two core Trithorax subunits, results in the downregulation of the same developmental genes that are also downregulated upon parp1 knockdown, demonstrating functional convergence on the same gene set. Second, our genome-wide analyses show that PARP1 binding occurs exclusively at loci enriched for mono-methylated histone marks and positive for Trithorax or Polycomb complexes. Finally, our biochemical assays demonstrate that PARP1 binds directly and with high affinity to mono-methylated histone marks in vitro. Together, these findings go beyond correlation and provide genetic, genomic, and biochemical support for the conclusion that PARP1 controls the expression of loci modified by Trithorax.

R2: Had the authors been able to perform in vivo experiments in Drosophila, the quality of the paper would have increased exponentially. However, if that is not possible, I suggest the following:

Fig. 1 D–H: I’d like to see the original data (the scan of the film), at least as a supplementary file.

Response: We have now included the original blot scan, both with and without the highlighted regions of interest (ROI) used for the analysis, as the new Supplemental Figure S1.

R2: Clearly indicate in the manuscript which NGS experiments are being re-analyzed, specifying the developmental stage of Drosophila in each case.

Response: We have revised the Materials and Methods section to clearly indicate that all NGS datasets were re-analyzed from raw data. Since all datasets except Trr, Brm, Iswi, and Mi-2 were generated from third instar larvae, we now specify only these exceptions in figure legends as originating from adult stages.

R2: The differences in binding and signal in ChIP-seq shown in Figures 2a, b, g, h, j, k; 3a, b, d; and 4a, b, e, f, g, h, i, j should be quantified using appropriate tools such as DiffBind. Qualitative plots are not sufficient, and the figures need to be completely restructured.

Response: We thank Reviewer 2 for this important comment. DiffBind is well suited for comparing the binding intensity of two different factors at the same loci, but it is not appropriate for assessing the distribution of a single factor across distinct groups of loci, which is the focus of our analysis. To address this concern, we have added three new supplemental figures (S3, S6, and S7) that provide quantitative boxplot analyses of factor occupancy across the different locus groups. These values were obtained using the multiBigwigSummary tool from the deepTools package. As these analyses are supportive, they are presented in the supplement, while the main figures continue to display the qualitative distribution profiles.

R2: It should be clearly stated that the findings are correlational and not causal, with a much more cautious tone, and that the conclusions are valid only in Drosophila.

Response: We would like to highlight that our discovery is supported not only by genome-wide correlations, but also by genetic knockdown experiments (Trr/Ash2 and parp1) and biochemical assays, which together strengthen the causal interpretation of PARP1 function at TrxG-modified loci.

R2: The tracks in Fig. 3e do not appear to be of good quality; a reference to the genomic location and signal intensity should be included.

Response: Figure 3E has been revised to include both genomic coordinates and signal intensity scales for all panels.

R2: For all analyzed NGS experiments, the replicates used must be specified.

Response: We have revised the Materials and Methods section to clearly specify the replicates used for each NGS dataset and to explain our selection criteria.

Round 2

Reviewer 1 Report

Comments and Suggestions for Authors

The authors have corrected all the shortcomings and responded to all the comments, the article can be published.

Author Response

We thank the reviewer for very helpful comments and suggestions.